# Polymyxin Resistance in *Salmonella*: Exploring Mutations and Genetic Determinants of Non-Human Isolates

**DOI:** 10.3390/antibiotics13020110

**Published:** 2024-01-23

**Authors:** Thais Vieira, Carla Adriana Dos Santos, Amanda Maria de Jesus Bertani, Gisele Lozano Costa, Karoline Rodrigues Campos, Cláudio Tavares Sacchi, Marcos Paulo Vieira Cunha, Eneas Carvalho, Alef Janguas da Costa, Jacqueline Boldrin de Paiva, Marcela da Silva Rubio, Carlos Henrique Camargo, Monique Ribeiro Tiba-Casas

**Affiliations:** 1Adolfo Lutz Institute, São Paulo 01246-000, SP, Brazil; tha-vieira@hotmail.com (T.V.); bio.cadrianas@gmail.com (C.A.D.S.); amandabertani94@gmail.com (A.M.d.J.B.); gisele.costa@ial.sp.gov.br (G.L.C.); karoline.campos@ial.sp.gov.br (K.R.C.); ctsacchi@gmail.com (C.T.S.); carlos.camargo@ial.sp.gov.br (C.H.C.); 2School of Veterinary Medicine, University of São Paulo, São Paulo 05508-270, SP, Brazil; cunha.mpv@gmail.com; 3Butantan Institute, São Paulo 05503-900, SP, Brazil; eneas.carvalho@butantan.gov.br (E.C.); alefjcosta@gmail.com (A.J.d.C.); 4R&D Department BioCamp Laboratories, Campinas 13082-020, SP, Brazil; jackboldrin@biocamp.com.br; 5School of Agriculture and Veterinarian Sciences, University of the State of São Paulo, Jaboticabal 14884-900, SP, Brazil; ma.rubio192@gmail.com

**Keywords:** *Salmonella* spp., polymyxin, colistin, antimicrobial resistance

## Abstract

Until 2015, polymyxin resistance was primarily attributed to chromosomal mutations. However, with the first report of mobile colistin resistance (*mcr-1*) in commensal *Escherichia coli* from food animals in China, the landscape has changed. To evaluate the presence of polymyxin resistance in *Salmonella* spp., a drop screening test for colistin and polymyxin B was carried out on 1156 isolates of non-human origin (animals, food, and the environment), received in Brazil, between 2016 and 2021. Subsequently, 210 isolates with resistant results in the drop test were subjected to the gold-standard test (broth microdilution) for both colistin and polymyxin B. Whole-genome sequencing (WGS) of 102 resistant isolates was performed for a comprehensive analysis of associated genes. Surprisingly, none of the isolates resistant to colistin in the drop test harbored any of the *mcr* variants (*mcr-1* to *mcr-10*). WGS identified that the most common mutations were found in *pmrA* (n= 22; T89S) and *pmrB* (n = 24; M15T, G73S, V74I, I83A, A111V). Other resistance determinants were also detected, such as the *aac*(*6*′)-*Iaa* gene in 72 isolates, while others carried beta-lactamase genes (*bla*_TEM-1_ *bla*_CTX-M-2_, *bla*_CMY-2_). Additionally, genes associated with fluoroquinolone resistance (*qnrB19*, *qnrS1*, *oqxA/B*) were detected in 11 isolates. Colistin and polymyxin B resistance were identified among *Salmonella* from non-human sources, but not associated with the *mcr* genes. Furthermore, the already-described mutations associated with polymyxin resistance were detected in only a small number of isolates, underscoring the need to explore and characterize unknown genes that contribute to resistance.

## 1. Introduction

Antibiotic use plays a significant role in the emerging public health crisis of antibiotic resistance in both human medicine and food production. Nevertheless, the veterinary use of antimicrobials is recognized as a key factor in the emergence of antimicrobial-resistant *Salmonella*. In addition to antimicrobial use in the veterinary field, co-resistance mechanisms should be taken into consideration in interpreting the spread and persistence of antimicrobial resistance [1,2,3]. Colistin serves as a last-resort drug to combat severe infections caused by multi-drug-resistant (MDR) and extensively drug-resistant (XDR) bacteria [4]. This underscores the urgent requirement for enhanced and effective continuous surveillance to control its dissemination.

In contrast to human medicine, colistin has been widely employed in veterinary medicine for many decades to treat and prevent infectious diseases. Despite efforts to minimize the inappropriate use of antibiotics in animals, colistin is still incorporated not only for preventing and treating infectious diseases but also for growth promotion, aiming to enhance performance, increase productivity, and prevent issues in the early weeks of life. Given that *Salmonella* spp. is a zoonotic pathogen, this practice facilitates and promotes the circulation of these resistance profiles between humans and animals [5,6].

Polymyxin resistance in Gram-negative bacteria primarily occurs dues to changes in the lipopolysaccharide (LPS) molecules that make up the outer layer of the outer membrane. In resistant strains, certain substances like 4-amino-4-deoxy-L-arabinose (L-Ara4N), phosphoethanolamine (PEtN), or galactosamine are added to the lipid A or the LPS core, reducing the negative charge of lipid A. In some isolates, the LPS part of the outer membrane may be entirely lost. These alterations are typically controlled by various two-component systems (TCSs) like PhoP/Q) and PmrA/B [7,8].

Since the first report of mobile colistin resistance (*mcr-1*) in commensal *Escherichia coli* from food animals in China [9], several studies have identified this gene and its variants in different species worldwide. To date, nine additional *mcr* gene groups (*mcr-2* to *mcr-10*) have been reported [10]. While the occurrence of *mcr* genes in *Salmonella* is rare in Brazil, we have detected colistin-resistant isolates in our laboratory’s routine antimicrobial resistance surveillance [11].

As part of continuous antimicrobial susceptibility surveillance, we aimed to characterize isolates of non-human origin regarding polymyxin resistance, identifying the presence of mutations in the chromosome or plasmid genes, in light of the emerging resistance to colistin.

## 2. Results

In this study, we performed a drop test screening of 1156 strains of *Salmonella* spp. isolates of non-human origin (from food, animals, and the environment) obtained between 2016 and 2021. This number corresponds to the total strains received in the laboratory during the study period (Appendix A). The most prevalent serotype was Heidelberg (14.7%), followed by Enteritidis (9%), Mbandaka (7.8%), and Typhimurium (7%) serotypes and its monophasic variant (6%).

Of the total of 1156 samples screened using the drop test, 210 isolates were found to be resistant to colistin, and among them, 175 isolates exhibited co-resistance to polymyxin B in the drop test screening (Table 1). We further examined the 35 isolates resistant to colistin but sensitive to polymyxin B in the drop test. Through the MIC test, we determined that 28 of these isolates were indeed resistant to colistin (MIC ≥ 4 mg/L), and among them, 14 were also resistant to polymyxin B. Consequently, out of the 35 isolates, only 7 were found to be sensitive to both drugs.

In Table 2, we present the overall MIC results for all 210 colistin-resistant isolates identified in the drop test. Among these 210 isolates, 16 had MIC values of ≤2 mg/L for colistin and were excluded from the further analysis. The MIC50 and MIC90, representing the concentrations inhibiting 50% and 90% of the population, were determined as 4 mg/L and 8 mg/L, respectively. All 210 isolates resistant to colistin in the drop test (n = 210) were found to be negative for the *mcr-1* to *mcr-10* genes by means of PCR.

Of the 210 colistin-resistant *Salmonella* isolates, 102 resistant isolates and 40 sensitive isolates (representative of the same serotypes, for comparison) underwent whole-genome sequencing to identify potential genes associated with antimicrobial resistance, enabling a more robust analysis of mutations.

In the analysis of 102 isolates (Appendix A), 26 had mutations in *pmrA*, 29 had mutations in *pmrB*, and 22 had mutations in both *pmrA/B* genes. Additionally, three isolates had mutations only in *phoP* (PhoP:p.A111V or phoP:p.N10K), while another three isolates exhibited mutations only in *phoQ* (PhoQ:p.T98A or phoQ:p.T61A). Among the 63 without mutations, 38 belonged to the O antigen group D1 (O9), all of which were of the Enteritidis serotype. The remaining 25 samples belonged to serogroups O4, O3,10, O7, O8, and O21 and did not exhibit mutations in the PmrA/B and PhoP/Q systems. Figure 1 illustrates the presence or absence of mutations in colistin-associated resistance genes for each isolate.

We sequenced 53 isolates from group D1 (O9). Among these, eight isolates with *pmrA* mutations were detected, and seven mutations were identified in *pmrB*, all within the O9 isolates. Among the three isolates that exhibited mutations in *phoP*, two isolates belonged to serogroup O9. Additionally, three isolates had mutations in *phoQ*, with two from serogroup O9. In all, 63 isolates showed no *pmrA/B* and *phoP/Q* mutations, with 38 isolates belonging to the D1 group and 25 belonging to serogroups O4, O3,10, O7, O8, and O21.

*S*. Enteritidis, the most frequent colistin-resistant serotype in our collection, tested negative for known *mcr*-genes. Subsequently, we analyzed mutations in other genes that have been previously explored for their association with colistin resistance, such as *pilN*, *ydeL*, *zraR*, *lolB*, *mdsC*, and *rfc*. Reference genes from the same serotype were also examined between sensitive and resistant isolates. Only a few mutations were identified, with two mutations (I98T and D108E) observed in the *pilN* gene on the Miami serotype, and one mutation (R247Q) observed in the *msdC* gene on the Anatum serotype. The *lolB* gene exhibited the E28K mutation in an *S*. Miami isolate.

The ResFinder tool, utilizing whole-genome sequencing (WGS), identified the mechanisms for antimicrobial resistance (AMR) present in all 102 isolates (Appendix A). The gene *aac*(*6*′)*-Iaa* was predominantly detected in the majority of the isolates, with 72 isolates carrying this aminoglycoside resistance gene. Notably, 68% of these isolates belonged to serogroup O9, primarily of the Enteritidis serotype. Seven isolates harbored the *bla*_TEM-1_ beta-lactamase gene, while one isolate exhibited the presence of the extended-spectrum beta-lactamase (ESBL) *bla*_CTXM-2_. Seventeen samples tested positive for *bla*_CMY-2_, an AmpC- type ESBL gene. Additionally, eleven isolates tested positive for genes associated with fluoroquinolone resistance, including *qnrB19*, *qnrS1*, and *oqxA/B*. Further, fifty-eight isolates showed mutations associated with quinolone and fluoroquinolone resistance, such as *gyrA* S83F, *gyrA* D87N, and *parC*:p.T57S. Different plasmid incompatibility groups were identified, with the main ones being IncC, IncF, colpVC, and Col(pHAD28).

## 3. Discussion

In this study, we evaluated the presence of polymyxin resistance in *Salmonella* from the food chain, and we examined the genetic determinants associated with this phenotype. Given the prevalence of animal sources such as poultry, chicken carcasses, and the farm environment (drag swab), our results underscore the importance of epidemiological monitoring. This is particularly crucial as the Enteritidis and Typhimurium serotypes are frequently associated with foodborne outbreaks worldwide and invasive infections [12,13]. Brazilian legislation has implemented control measures in commercial poultry establishments for broiler chickens and turkeys, aiming to reduce the prevalence of *Salmonella*, especially of the Enteritidis and Typhimurium serotypes, to establish an adequate level of protection for consumers [14].

The drop test has been employed as an alternative to enhance the accuracy of colistin resistance screening. This test is capable of detecting colistin resistance mediated by mutations in the chromosome and by *mcr* producers [15,16]. However, its use is not recommended as a standard method due to its error rate for polymyxin elution, likely stemming from differential binding to micropipette tips or a distinct diffusion gradient around the dripped drop [17]. Given that the emergence of colistin resistance is a recent global phenomenon, implementing rapid and reliable screening tools to detect and analyze colistin-resistant pathogens for tailored patient care is imperative. Furthermore, the phenomenon of heteroresistance to colistin is largely underestimated [18].

We decided to use the drop test to screen a large number (>1000) of isolates, and 83% of the resistant isolates were confirmed by means of the gold-standard method (broth microdilution). These results could be beneficial in routine antimicrobial resistance monitoring in the laboratory, considering the high demand for isolates and the necessity to monitor the presence of isolates resistant to these drugs, which are used as a last resort in patient treatment [4].

We are aware that one limitation to be addressed is that a validation of the drop test’s effectiveness was not explicitly conducted in this study. MIC testing was performed only on strains identified as resistant in the drop test, and the method was exclusively utilized for screening purposes. Nevertheless, a comparison of our results with findings from the literature suggests efficacy. For instance, Conceição-Neto et al. (2020) [18] reported a sensitivity of 74% for the drop test in identifying resistant *K. pneumoniae*, while Jouy et al. (2017) [15] achieved 100% identification of resistant *E. coli* isolates using the drop test.

Unexpectedly, no plasmid resistance gene (*mcr)* was detected. While there are reports of *mcr* gene circulation in *E. coli* and *Salmonella* spp. in Brazil, data remain scarce [18,19,20,21]. As a reference laboratory, the Adolfo Lutz Institute (IAL) receives *Salmonella* spp. from different sources, contributing to the understanding of polymyxin resistance mechanisms. In 2020, we detected the first strain of *Salmonella* positive for the *mcr-1* gene in a strain of human origin, which alerted us to the possible dispersion of this gene in the environment and among animals [11]. More recently, we detected the *mcr-9* gene in a polymyxin-sensitive *Salmonella* strain of animal origin [22]. Studies have indicated that the *mcr-9* variant may be silently spreading since the resistance phenotype is not expressed [23,24].

In Brazil, the Ministry of Agriculture, Livestock and Food Supply (MAPA) published *Normative Instruction* [25], which prohibits the importation and manufacture of the antimicrobial substance colistin sulfate for use as a performance-enhancing zootechnical additive in animal feed throughout the national territory. China has also banned the use of colistin in agriculture. However, despite these bans on colistin use, studies suggest that the spread of *mcr-1* in hospital environments and communities may continue. This raises questions about the effectiveness of these bans in containing the spread of *mcr-1* [26].

Specifically, our analysis revealed only three distinct point mutations in *phoP* and *phoQ.* In line with expectations for the *Salmonella* genus, where colistin resistance is commonly associated with *pmrA/B* gene mutation [27,28], our study identified mutations such as T89S in *pmrA* and G73S, V74I, I83V, and A111T in *pmrB.* While these mutations have been previously associated with colistin resistance in the literature, none were predicted to affect protein function [29,30,31].

However, we did not determine the individual contribution and cumulative effects of these genes on colistin resistance. Further investigations, such as complementation tests or site-directed mutagenesis, would be necessary. It is also worth noting that missense mutations may not necessarily result in increased minimum inhibitory concentrations (MICs) for colistin [28].

In our study, among the 63 isolates resistant to colistin without mutations in the investigated genes, 38 were identified as part of the O antigen of the D1 group (O9) and belonged to the Enteritidis serotype. These findings raise concerns, particularly due to the prevalence of *S*. *enteritidis* in human infections associated with foodborne outbreaks and gastrointestinal illnesses worldwide. In 2012, a study recommended evaluating colistin-resistant isolates at the serotype level [28].

In *Salmonella* group D1, including *S. enterica* serovar Enteritidis, intrinsic resistance to colistin is linked to the somatic antigen epitope (O). Unlike group B, group D1 has tyvellosis instead of abequosis as the O antigen side-branch sugar. Increased susceptibility to colistin in group D *Salmonella* may result from a frameshift mutation in the *rfc* gene, encoding the O antigen polymerase. This mutation leads to truncated RFC protein, causing inefficient assembly and polymerization of O antigen subunits, resulting in a rougher LPS and a more permeable cell membrane. Despite structural similarities between abequose and tyvellosis, subtle differences in the hydroxyl group’s position may reduce susceptibility to colistin across different serotypes. Tyvellosis may prevent colistin from reaching its target, the LPS component of the bacterial outer membrane, the initial cellular target of polymyxin [32,33].

In our study, even when comparing sensitive and resistant isolates within the same serotype, we did not detect known and/or unknown mutations in the studied genes that could confer resistance to polymyxins. The large proportion of isolates with unknown resistance mechanisms in the current study indicates that other, yet uncharacterized, resistance mechanisms may be more important for *Salmonella* spp.

In our study, 56 samples from serogroups O4, O3,10, O7, O8, and O21 did not exhibit mutations in the PmrA/B and PhoP/Q systems, nor did they show the presence of plasmid resistance genes. These findings suggest that, even in the absence of intrinsic resistance linked to the O antigen and the absence of mutations in the two-component system, another, as yet uncharacterized, mechanism may be causing the observed resistance.

Resistance to polymyxin antibiotics relies on an intricate mechanism involving multiple genes that affect the cell surface, compromising its integrity or leading to modifications [32,33]. The identification of plasmid-mediated mechanisms of resistance to polymyxins prompts us to consider the perspective and assess the extent of the dissemination of this resistance in both human and veterinary medicine, along with understanding its impact.

Despite extensive efforts to elucidate the mechanisms behind polymyxin resistance, a significant amount of information remains undiscovered, as there are resistant strains for which the underlying resistance mechanisms are still unknown. Emphasizing the detection of polymyxin-resistant isolates is crucial. Both retrospective and prospective epidemiological surveys are necessary, given the current scarcity of knowledge on this issue.

## 4. Materials and Methods

Bacterial isolates

In this study, we analyzed 1156 *Salmonella* isolates from non-human sources, received from 2016 to 2021 for serotyping at Institute Adolfo Lutz, a reference laboratory for public health in Brazil. These isolates had been isolated and presumptively identified in animal pathology and food microbiology laboratories in various geographic locations of the State of São Paulo.

The 1156 *Salmonella* isolates represented non-human sources, with isolates from animals (557), primarily from poultry; food (363), including food-producing animals and other foodstuffs; and the environment (236), mainly from sewage sludge and poultry environments. For this study, the entire set of isolates of non-human origin received at the Adolfo Lutz Institute from 2016 to 2021 was utilized.

Firstly, all isolates underwent confirmation as members of the *Salmonella* genus though conventional biochemical tests. The determination of subspecies was conducted based on additional biochemical characteristics [34]. *Salmonella* serotyping was performed according to the 9th edition of the White–Kauffmann–Le Minor scheme on the basis of somatic O and H flagellar antigens by agglutination tests with antisera [34] (prepared in the Laboratory of Enteric Pathogens, Adolfo Lutz Institute, São Paulo).

Colistin drop test

The testing method described by Jouy et al. (2017) for *E. coli* [15] and by Pasteran et al. (2018) [16] was employed in the present work. Specifically, a single 10 µL drop of a 16 mg/L colistin solution was deposited on an MHA plate (Oxoid) previously swabbed with a 0.5 McFarland suspension of the bacterial isolate. The plates with the solution were allowed to sit for 15 min at room temperature (ensuring complete absorption of the drop before moving the plate), then inverted and incubated for 16 to 18 h at 35 °C. After incubation, the presence or absence of an inhibitory zone was determined with transmitted light, and halos were recorded for standardization purposes.

An isolate was classified as colistin-susceptible if a clear inhibition zone was observed, regardless of the diameter. An isolate was called colistin-resistant if there was an absence of an inhibition zone or if defined colonies within the inhibition zone, indicative of heteroresistant subpopulations, were observed. To validate the results of antimicrobial susceptibility testing (AST), quality-control strains, namely, *E. coli* ATCC 25922 and *P. aeruginosa* ATCC 27853, were employed.

Minimum inhibitory concentrations

MIC values for colistin were obtained using the recommended broth microdilution (BMD) method, following EUCAST/BrCAST guidelines [35], employing an in-house microdilution technique in cation-adjusted Mueller–Hinton broth (CA-MHB). Colistin sulfate powder (Sigma, St. Louis, MI, USA) was dissolved in CA-MHB (Oxoid) (stock solution, 128 mg/L), and log2 dilutions were subsequently prepared to achieve an MIC range of 0.5 to 64 mg/L.

Strains were considered to have acquired resistance to colistin when the MIC was higher than 4 mg/L, in accordance with EUCAST standards (35). Quality-control strains *E. coli* ATCC 25922 and *P. aeruginosa* ATCC 27853 were utilized to validate the results of AST.

PCR amplification

Total DNA extraction from overnight cultures of *Salmonella* isolates was performed using the Wizard Genomic DNA purification System (Promega, Madison, WI, USA). The screening for *mcr-1* to *mcr-10* was carried out through a multiplex PCR protocol [36,37].

DNA Extraction, Whole-Genome Sequencing, and Assembly

The entire bacterial DNA from the 142 isolates (comprising 102 resistant and 40 susceptible isolates) was extracted following the manufacturer’s instructions using the Wizard Genomic DNA Purification kit (Promega) designed for bacterial cultures. Following extraction, DNA quantification was performed utilizing a Qubit fluorometer (Thermo Scientific Inc., Waltham, MA, USA). Subsequently, libraries were prepared for Illumina NextSeq sequencing using a P1/300 cycle cartridge. Both the library preparation and Illumina runs were conducted at the Strategic Laboratory, Instituto Adolfo Lutz, São Paulo, Brazil. Genomes were de novo assembled using CLC Workbench 11 software (Qiagen, Hilden, Germany).

Annotation, Resistome and Virulome Detection, Serotype Prediction, and MLST

The assembled genomes were uploaded to the Galaxy Europe platform [38] and subsequently annotated using Prokka. The identification of acquired resistance and virulence-codifying genes was accomplished through Abricate, utilizing Resfinder, accessed on 10 October 2023. Chromosomal mutations linked to antimicrobial resistance were identified using Bionumerics version 8.1 software (Applied Maths, Sint-Martens-Latem, Belgium) and Resfinder, available on the Center for Genomic Epidemiology webserver [39] (https://www.genomicepidemiology.org/, accessed on 10 October 2023).

The in silico serotype was determined using the Seqsero program, available on the Center for Genomic Epidemiology webserver [39] (https://www.genomicepidemiology.org/, accessed on 15 July 2023). Sequence types (STs) were assigned based on the internal sequences of seven housekeeping genes, available on the PubMLST webserver (https://pubmlst.org/, accessed on 20 July 2023). The heatmap and its hierarchical clusters were created in R (version 4.3.1; R Foundation, Vienna, Austria), using the function heatmap.2 from the gplots package. All the sequences generated in this study were deposited in the GenBank database under BioProject: PRJNA1049413. 

## Figures and Tables

**Figure 1 antibiotics-13-00110-f001:**
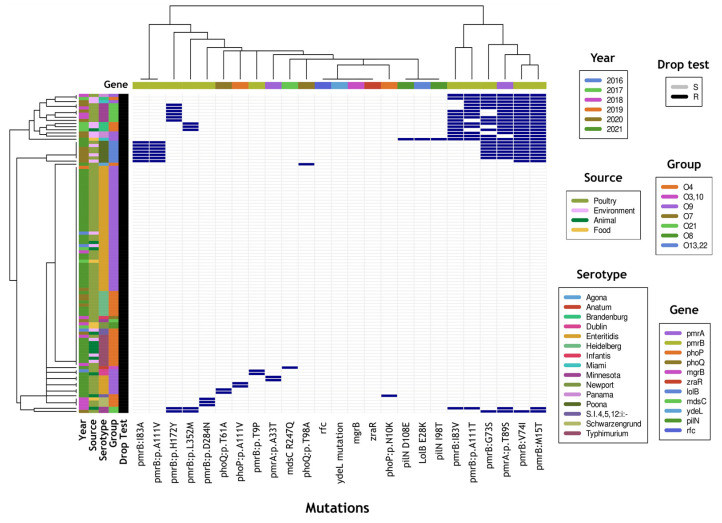
Heatmap visualization of the presence or absence of mutations in colistin-associated resistance genes in *Salmonella enterica* isolates. Vertical bar colors represent the year, source, serotype, O antigen group, and drop test results; horizontal bar colors represent the gene in which each mutation was detected. Dark blue color represents the presence of a mutation. A hierarchical cluster based on the pattern of presence/absence is shown for genomes (left) and for genes (top). The database used to build this figure is presented in Appendix A.

**Table 1 antibiotics-13-00110-t001:** Number of isolates resistant to colistin (drop test) in the period 2016–2021.

Serotype	Number of Isolates Resistant to Colistin (Drop Test)						
	2016	2017	2018	2019	2020	2021	Total
Agona	0	0	0	1	0	1	2
Brandenburg	0	5	0	0	0	0	5
Dublin	3	0	0	0	0	1	4
Enteritidis	43	3	1	0	1	34	82
Gallinarum	0	2	0	1	2	4	9
Grumpensis	0	0	0	1	0	0	1
Heidelberg	0	1	11	0	3	5	20
Infantis	0	0	1	0	0	0	1
Javiana	3	0	0	0	0	0	3
Madelia	0	0	0	1	0	0	1
Mbandaka	1	0	0	0	0	0	1
Miami	4	0	0	0	0	1	5
Minnesota	0	1	6	0	8	0	15
Newport	0	1	0	0	0	0	1
Ohio	0	0	0	0	0	2	2
Panama	0	0	1	0	1	0	2
Poona	0	0	0	2	9	2	13
Pullorum	0	1	0	0	1	3	5
Rubislaw	0	0	1	0	0	0	1
*S.enterica* subsp.*enterica*	0	0	1	0	1	2	4
*S*. I. 4,[5],12:i:-	12	0	1	0	1	1	15
Sandiego	0	1	0	0	0	0	1
Schwarzengrund	1	0	3	0	0	0	4
Typhimurium	0	0	2	0	0	10	12
Winslow	0	0	0	0	0	1	1
Total	67	15	28	6	27	67	210

**Table 2 antibiotics-13-00110-t002:** Results of minimum inhibitory concentrations obtained via broth microdilution of all 210 colistin-resistant isolates in the drop test.

Serotype	Colistin (mg/L)						
	0.5	1	2	4	8	16	Total
Agona	0	0	0	2	0	0	2
Anatum	0	0	0	1	0	0	1
Brandenburg	0	0	0	3	2	0	5
Dublin	0	0	0	3	1	0	4
Enteritidis	0	0	0	41	32	12	85
Gallinarum	0	0	0	1	3	2	6
Grumpensis	0	0	0	0	1	0	1
Heidelberg	0	0	2	12	4	3	21
Infantis	0	0	0	1	0	0	1
Javiana	1	1	0	0	1	0	3
Madelia	0	0	0	1	0	0	1
Mbandaka	0	0	0	1	0	0	1
Miami	0	0	0	5	0	0	5
Minnesota	0	0	3	11	0	1	15
Newport	0	0	0	1	0	1	2
Ohio	0	0	0	1	1	0	2
Panama	0	0	0	2	0	0	2
Poona	0	0	3	11	0	0	14
Pullorum	0	0	0	2	3	0	5
Rubislaw	0	0	0	0	1	0	1
*S*. I. 4,[5],12:i:-	0	1	1	11	1	0	14
Sandiego	0	0	0	0	1	0	1
Schwarzengrund	0	1	0	1	1	1	4
Typhimurium	0	0	2	10	0	1	13
Winslow	0	0	1	0	0	0	1
Total	1	3	12	121	52	21	210

## Data Availability

All the sequences generated in this study were deposited in the GenBank (BioProject: PRJNA1049413).

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
