# Peer review of "Polymyxin Resistance in Salmonella: Exploring Mutations and Genetic Determinants of Non-Human Isolates"

_antibiotics, 2024, doi:10.3390/antibiotics13020110_

Round 1

Reviewer 1 Report

Comments and Suggestions for Authors

In this study, the authors analyzed the genetic resistance in polymyxin-resistant Salmonella recovered from non-human samples in Brazil between 2016 and 2021. Among 1156 analyzed Salmonella isolates, 210 showed phenotypic resistance by using drop method, from them 16 isolates showed a MIC value less than 2µg/mL and were considered susceptible by broth microdilution assay. Multiplex PCR showed the absence of mcr genes in the resistant isolates. However, whole genome sequencing analysis showed the presence of chromosomic mutations encoding resistance to polymyxin. Nevertheless, 63 resistant isolates did not show any mutation mechanism and may resist polymyxin by different or not understood mechanisms. I found this study is timely and relevant to the broad readership of the journal. However, some revisions should be addressed before considering this manuscript for publication.

- Try to reduce the abstract to about 200 words.

Line 23 “received between 2016 and 2021 », where. Kindly add the name of the country where this study was carried out “Brazil”.

- Lines 46-48: “Nevertheless, the veterinary use of antimicrobials is recognized as a key factor in the emergence of antimicrobial-resistant Salmonella”. In addition to antimicrobial use in the veterinary field, co-resistance mechanisms should be taken into consideration in interpreting the spread and persistence of antimicrobial resistance (Doi: 10.1016/j.jhazmat.2022.129476; Doi: 10.1016/j.chemosphere.2018.10.114).

- What is the purpose of sequencing the 40 susceptible isolates? This number can be taken among the resistant isolates to boost further the analysis with more resistant isolates.

- Line 130: correct “We sequenced 53 isolates were from group D1 (O9)”.

- The sequences generated in this study are deposited under BioProject PRJNA1048438, the date is not yet available on the database.

- Lines 312-314: Do the authors use a positive control when detecting mcr genes by Multiplex PCR?

- Among the 210 resistant Salmonella by drop method, 16 showed a MIC value less than 2 µg/mL, as microdilution is considered the “standard method” to assess resistance toward polymyxin, these isolates should be considered susceptible and excluded from the genomic analysis.

- Among 210 resistant isolates, 102 were considered for WGS, is there any criteria for this selection?

- For more interpretation of results, authors should analyze the obtained sequences for the presence of plasmids that may carry some transferable resistance genes.

Author Response

Thank you for considering our study for publication in Antibiotics. We received the reviewers’ comments and we carefully read the considerations; in order to improve the manuscript quality, we clarify each point outlined by the reviewers, as follows:

1Try to reduce the abstract to about 200 words.

The abstract has been condensed.

2-Line 23 “received between 2016 and 2021 », where. Kindly add the name of the country where this study was carried out “Brazil”.

Now we included the country in the text.

3-Lines 46-48: “Nevertheless, the veterinary use of antimicrobials is recognized as a key factor in the emergence of antimicrobial-resistant Salmonella”. In addition to antimicrobial use in the veterinary field, co-resistance mechanisms should be taken into consideration in interpreting the spread and persistence of antimicrobial resistance (Doi: 10.1016/j.jhazmat.2022.129476; Doi: 10.1016/j.chemosphere.2018.10.114).

The text was rewritten following the reviewer's suggestion, and the reference was added.

4-What is the purpose of sequencing the 40 susceptible isolates? This number can be taken among the resistant isolates to boost further the analysis with more resistant isolate

Because recent studies on polymyxin resistance in the Salmonella genus indicate the necessity of conducting a comparative analysis of mutations among serotypes and between susceptible and resistant strains.

5-Line 130: correct “We sequenced 53 isolates were from group D1 (O9)”.

The text has been corrected.

6-The sequences generated in this study are deposited under BioProject PRJNA1048438, the date is not yet available on the database.

The data has already been processed and will be released upon publication of the article.

-7-Lines 312-314: Do the authors use a positive control when detecting mcr genes by Multiplex PCR?

Yes, we have the positive controls for mcr genes.

8-Among the 210 resistant Salmonella by drop method, 16 showed a MIC value less than 2 µg/mL, as microdilution is considered the “standard method” to assess resistance toward polymyxin, these isolates should be considered susceptible and excluded from the genomic analysis.

Yes, these isolates were disregarded, and we worked only with the resistant isolates through the standard method.

9-Among 210 resistant isolates, 102 were considered for WGS, is there any criteria for this selection?

We sequenced the majority of isolates that were not part of the O9 group precisely because it is mentioned in the literature that there may be resistance linked to the serotype. And we also selected O9 isolates to identify possible mutations associated with colistin resistance

10-For more interpretation of results, authors should analyze the obtained sequences for the presence of plasmids that may carry some transferable resistance genes.

Thank you for the correction of the article and the suggestions, which were accepted. Some more robust analyses will be included in another article

Reviewer 2 Report

Comments and Suggestions for Authors

Polymyxin Resistance in Salmonella: Exploring Mutations and  Genetic Determinants of Non-Human Isolates

Authors:  Thais Vieira, Carla Adriana dos Santos, Amanda Maria de Jesus Bertani, Gisele Lozano Costa, Karoline  Rodrigues Campos, Cláudio Tavares Sacchi, Marcos Paulo Vieira Cunha, Eneas Carvalho, Alef Janguas, Jacqueline Boldrin de Paiva, Marcela da Silva Rubio, Carlos Henrique Camargo, Monique Ribeiro Tiba-Casas

The objective of this study aimed to characterize isolates of non-human origin regarding polymyxin resistance, identifying the presence of mutations in the chromosome or plasmid genes, in light of the emerging resistance to colistin.

General comments: This article is interesting on the fact that the diffusion of mcr genes seems to be weak or even non-existent. However, the strategy implemented is incomplete from a genetic point of view. It would have been desirable to search for mutations in other 2-component systems such as CcrAB. Furthermore, the involvement of overexpression of the acrAB_tolC system or pmrD is not even discussed as a hypothesis for increasing colistin MICs.

Major comments:

-          Lines 223-250:

o   In K. pneumoniae, mutations in the TCS CrrAB upregulates the expression of CrrC,

resulting in CST resistance : https://doi.org/10.1128/AAC.00009-16.

o   In E. cloacae complex, the study of Pantel & al revealed that in the C-XI strain ECL13047, the TCS ECL_01761-ECL_01762 (ortholog of CrrAB from K. pneumoniae) is involved in CST resistance resulting in CST resistance. Doi 10.1128/aac.00776-22.

o   Similarly, ECL_01760, an ortholog of CrrC, contributes to CST hetero-resistance in ECL13047.

o   pmrD is also involved in colistin resistance. Overexpression of PmrD alone in Salmonella Typhimurium is sufficient to confer colistin resistance. PmrD is involved in the regulation of the PmrAB system via binding to PmrA, leading to persistent expression of the PmrA-activated genes in S Typhimurium, in which it connects the PhoPQ and PmrAB systems. This point is not discussed for strains without mutations in the targeted genes? https://doi.org/10.1016/S1473-3099(16)30394-2

o   AcrAB efflux pump was also involved in colistin resistance. This point is not discussed for strains without mutations in the targeted genes? https://doi.org/10.3389/fmicb.2023.1207441.

Minor comments:

-          please replace throughout the text: µg/ml by mg/L

Author Response

Reviewer 2

I would like to express our sincere gratitude for accepting to review and consider our article for publication. Furthermore, your time and expertise are greatly appreciated.

This article is interesting on the fact that the diffusion of mcr genes seems to be weak or even non-existent. However, the strategy implemented is incomplete from a genetic point of view. It would have been desirable to search for mutations in other 2-component systems such as CcrAB. Furthermore, the involvement of overexpression of the acrAB_tolC system or pmrD is not even discussed as a hypothesis for increasing colistin MICs

 In terms of the genetic perspective, we agree with the reviewer's observation regarding the omission of certain genes in the assessment. However, our study deliberately concentrated on specific genes known to be linked to polymyxin resistance (pmrA/B, phoP/Q). Additionally, we explored genes associated with the O9 antigen (pilN, ydeL, zraR, lolB, mdsC, and rfc), as well as genetic determinants (mcr). It's worth noting that we are currently engaged in another project where we will incorporate additional potential resistance genes, specifically pmrD and the acrAB_tolC system.

Minor comments:

-          please replace throughout the text: µg/ml by mg/L

The exchange has been made

Round 2

Reviewer 2 Report

Comments and Suggestions for Authors

Polymyxin Resistance in Salmonella: Exploring Mutations and  Genetic Determinants of Non-Human Isolates

Authors:  Thais Vieira, Carla Adriana dos Santos, Amanda Maria de Jesus Bertani, Gisele Lozano Costa, Karoline  Rodrigues Campos, Cláudio Tavares Sacchi, Marcos Paulo Vieira Cunha, Eneas Carvalho, Alef Janguas, Jacqueline Boldrin de Paiva, Marcela da Silva Rubio, Carlos Henrique Camargo, Monique Ribeiro Tiba-Casas

The objective of this study aimed to characterize isolates of non-human origin regarding polymyxin resistance, identifying the presence of mutations in the chromosome or plasmid genes, in light of the emerging resistance to colistin.

-          Thank you for responding to my suggestions

-          The answers to my questions are not satisfactory.

-          I'm sorry, but within the limits of the study, no hypothesis of uncharacterized mechanisms involved in colistin resistance is discussed.

Without this comprehensive scientific approach, it would be necessary to submit this article as a letter!

Author Response

Thank you for the careful evaluation of our study and the suggestions. 
We are presenting the results obtained using Bionumerics, focusing on the analysis of mutations in colistin resistance-associated genes pmrD and the efflux pump system acrB. In line with our findings in other genes, no mutations were detected in the regulatory gene pmrD. Our analysis involved extracting the gene from each studied genome, comparing isolates within the same serotype, and distinguishing between sensitive and resistant strains.
Despite the well-established association between specific mutations in two-component systems PmrAB and PhoPQ, along with their regulator PmrD, and colistin resistance in Enterobacteriaceae, including Klebsiella pneumoniae and Salmonella Enterica, as well as A. baumannii, our study did not identify any mutations in pmrD. Regarding the potential overexpression of the acrAB_tolC system or pmrD, we did not conduct experiments to verify this aspect.
We acknowledge and agree to the possibility of the article being published as a letter. 
Attached is a screenshot of the Bionumerics software interface for your reference.

Round 3

Reviewer 2 Report

Comments and Suggestions for Authors

Answers to questions are satisfactory